# Neural network-based ensemble approach for multi-view facial expression recognition

**Muhammad Faheem Altaf[1], Muhammad Waseem Iqbal[2], Ghulam Ali[3], Khlood Shinan[4], Hanan E. Alhazmi[5], Fatmah Alanazi[6], M. Usman Ashraf[7]***

1 Department of Computer Science, Superior University Lahore, Lahore, Pakistan, 2 Department of Software Engineering, Superior University Lahore, Lahore, Pakistan, 3 Department of Software Engineering, University of Okara, Okara, Pakistan, 4 Department of Computers, College of Engineering and Computers in Al-Lith, Umm Al-Qura University, Makkah, Saudi Arabia, 5 Computer Science Department, College of Computer and Information Systems, Umm Al-Qura University, Makkah, Saudi Arabia, 6 Computer Science Department, College of Computer and Information Sciences, Imam Muhammad Bin Saud University, Riyadh, Saudi Arabia, 7 Department of Computer Science, GC Women University Sialkot, Pakistan

* usman.ashraf@gcwus.edu.pk

## Abstract

In this paper, we developed a pose-aware facial expression recognition technique. The proposed technique employed K nearest neighbor for pose detection and a neural network-based extended stacking ensemble model for pose-aware facial expression recognition. For pose-aware facial expression classification, we have extended the stacking ensemble technique from a two-level ensemble model to three-level ensemble model: base-level, meta-level and predictor. The base-level classifier is the binary neural network. The meta-level classifier is a pool of binary neural networks. The outputs of binary neural networks are combined using probability distribution to build the neural network ensemble. A pool of neural network ensembles is trained to learn the similarity between multi-pose facial expressions, where each neural network ensemble represents the presence or absence of a facial expression. The predictor is the Naive Bayes classifier, it takes the binary output of stacked neural network ensembles and classifies the unknown facial image as one of the facial expressions. The facial concentration region was detected using the Voila-Jones face detector. The Radboud faces database was used for stacked ensembles' training and testing purpose. The experimental results demonstrate that the proposed technique achieved 90% accuracy using Eigen features with 160 stacked neural network ensembles and Naive Bayes classifier. It demonstrates that the proposed techniques performed significantly as compare to state of the art pose-ware facial expression recognition techniques.

## 1. Introduction

The research of automatic recognition of pose-aware facial expression has endorsed significant progress in the past [1,2]. Automatic recognition of eight basic expressions (neutral, anger, happiness, surprise, fear, sadness, disgust, and contempt) from frontal pose has been done with fairly high accuracy. However, recognition of multi-pose facial expressions is still

**Data availability statement:** The data underlying the results presented in this study are available via RaFD after receiving the proper approvals. Requests to access the dataset can be sent to info@rafd.nl.

**Funding:** The author(s) received no specific funding for this work.

**Competing interests:** The authors have declared that no competing interests exist.

a challenging problem, primarily due to the dissimilarity in multi-pose facial expression representation. Most of the existing techniques make use of images that are relatively still and demonstrate posed facial expressions in a near frontal pose [2]. Therefore, several real-life applications relate to multi-pose facial expressions in human-to-human interaction, where the conjecture of having immovable subjects is impractical.

However, extracting facial features from facial images independent of pose is a difficult task because any change in poses also affects the facial expression representation. Consequently, multi-pose facial expression classification requires a large amount of training data to learn variations among facial expressions and poses, which are generally not available [3]. In addition, the accuracy of the classifier decreases when applied on facial images with continuous changes in pose. Multi-pose facial expression recognition techniques rely on 2D/3D facial images to distinguish the image variation caused by change in facial expression and pose. However, the accuracy of multi-pose expression recognition depends on the accuracy of pose detection, which is not an easy task [4].

According to Ekman, the facial region around the eyes and mouth contains more information about action units as compare to other regions of face [5]. This suspicion motivated us to focus on extraction of facial concentration region for facial feature representation. We have utilized the Voila-Jones face detection method [6] to find and clip the facial region from the facial picture in order to do this. The identified face concentration region's size, contrast, and brightness have all been adjusted to be normal. Subsequently, we used histogram of oriented gradient (HOG) and principal component analysis (PCA) for feature extraction. These features were mapped to the sample image into a feature space where stacked neural network ensemble (SNNE) is used to recognize the facial expression. Another interesting point is that posed facial image provides partial facial information. Consequently, the facial expression information entailed in the facial images related to multiple poses is not same. This situation inspires us to detect the pose before applying the facial expression classification techniques. As a result, we propose a pose-aware facial expression recognition technique, for pose detection and pose specific expression recognition. For pose-aware facial expression classification we have extended the stacking ensemble technique from two-level ensemble model to three-level ensemble model: base-level, meta-level and predictor. The base-level classifier is the binary neural network. The meta-level classifier is a pool of binary neural networks. The outputs of binary neural networks are combined using probability distribution from equation 1 and 2 to build the neural network ensemble as presented in [7]. The output of a neural network ensemble is either a one or a zero, which indicates whether an expression is present or not. Each neural network ensemble in the pool of SNNEs is trained to represent the presence or absence of a facial expression in order to understand the similarities across multi-pose facial expressions. The Naive Bayes (NB) classifier, which draws its inspiration from, is the predictor [8]. It identifies the sample expression as one of the possible outcomes using the binary output of SNNEs. The reason behind using NB is that it operates over binary data, it either accepts or rejects the existence of an expression. Moreover, the K nearest neighbor (KNN) classifier is used for determining the relationship between the facial features of multiple poses. In the KNN classifier, the response is the detection of pose from corresponding facial feature vector of a given facial image.

The major objectives of this research were to develop a pose-aware facial expression recognition technique while satisfying the following conditions.

- The binary neural networks were trained and tested on multi-pose dataset.

- Extraction of features which are invariant to differences of facial structure and head pose.

- The availability of sufficient multi-pose dataset.

- The system works automatically without human intervention.

Rest of the paper organized as follows: The related work presented in section 2. Methodology, framework implementation, examination of the restrictions, constraints, and design choices indicated in section 3. Section presents the experimental findings. Finally, part 5 and section 6 separately give the discussion and conclusion.

## 2. Related work

Until early 2008, the issue of pose-aware facial expression recognition was comparatively ignored in the literature. Hu et al. have pointed out this issue in [3]. In the facial expression recognition literature, the use of multi-pose facial images is rare; and comparatively less attention has been given to the problem of multi-pose facial expression recognition. Hu et al. [3] made the first attempt to recognize the multi-pose facial expressions. The facial features around eye brow, eyes and mouth were detected using 2D displacements of 38 facial landmark points. The linear Bayes, quadratic Bayes, parzen classifier, support vector machine (SVM) and KNN classifiers employed on facial land marks to evaluate the performance of multi-pose facial expression classification techniques. Achieved best average recognition accuracy on BU-3DFE facial expression database at the 45-degree facial pose. Ognjen et al. [9] proposed the coupled scaled gaussian process model to normalize the pose for multi-pose facial expression recognition. The model learns the relationship between each pair of poses to determine the dependencies among multiple poses.

Most of the facial expression recognition systems employed the Ekman's [5,10] facial action coding system to represent the facial expression in the form of action units. This technique involves a lot of manual effort to label the images as mentioned by Chew et al. [11]. Many newly developed techniques such as [12] and [13] employed facial action coding system, were unable to define the relationship between action units and facial expression recognition techniques. This issue leads towards the use of appearance-based feature extraction techniques.

Mostafa et al. [2] used the local binary pattern (LBP), Sobel and discrete Laplace features to recognize the spontaneous facial expressions with multiple poses (0, 18, 36, 49, 62, 75 and 90). Proposed a random forest based novel ensemble classifier to classify the multi-pose facial expressions using different datasets for the training and testing of classifiers. The objects of the facial expression dataset belong to different cultural, ethnic, and geographical regions. The best average accuracy was obtained on frontal pose facial expressions. Zheng et al. [14] used the scale invariant feature transformation (SIFT) feature vector to recognize the multi-pose facial expressions. Similarly, in [4] facial landmarks extracted from a set of SIFT features to represent the five poses for multi-pose facial expression recognition. The experimental results show that the best average expression recognition accuracy achieved with 45-degree pose. Moore and Bowden [15] proposed a multi class SVM classifier for multi-pose facial expression recognition using texture descriptor. Various variants of LBP descriptor were used for feature extraction to investigate the effect of orientation and multi-resolution analysis for non-frontal facial expression recognition. The sample image divided into a set of grids to extract the LBP features. The LBP features of each grid were then finally concatenated to form a feature vector to represent the corresponding facial image. The experiments were performed on BU-3DFE and multi-PIE databases to evaluate the optimal poses of multi-pose facial expressions.

Recently, Wenming [16] developed the group sparse reduced-rank regression model for multi-pose facial expression recognition. The model describes the correlation between facial features and the subsequent expression class label vector by selecting the optimal facial regions, that contribute most to the expression recognition. Each sample facial

image divided into a set of equal size facial regions. The facial feature vector was extracted using LBP descriptor. A comprehensive review on using deep learning techniques for facial expression is presented in [17]. Karnati et al. [18] introduced a parallel network based on texture features to overcome the issue of intra class facial appearance variation in representing the facial expressions. Karnati et al. [19] developed a novel technique for illumination normalization to recognize the facial expression in wild, the main focus was on the extraction of five prominent facial regions that contribute most in the representation of facial expression variations. More recently a hybrid deep convolutional neural network-based technique developed to efficiently recognize the facial expressions, where the geometric features extracted using local gravitational force descriptor and the holistic features extracted using the convolution layer [20].

In this work, we designed a novel ensemble model using Radboud faces database (RaFD). The proposed ensemble model permits the SNNEs to represent the multiple binary neural networks by a singular value, that is the usual case as presented in ensemble classifiers literature [21,22]. Thus, Each SNNE's binary values signify if an expression is present. The amount of binary neural networks in each SNNE, which is substantial, regulates the difficulty of multi-pose expression recognition.

## 3. Contributions

The contributions of this research are as follow:

1. Novel Ensemble Classifier: The proposed neural network-based ensemble model is a novel ensemble structure that is the extended version of stacking ensemble model.

2. Pose-aware facial expression recognition: We developed a two-level approach to recognize the pose-aware facial expression recognition: at first level it detects the pose from a given facial image then invokes the pose specific classifier to recognize pose-aware facial expression. The use multiple pose-specific models quantify the multi-pose complex problem to a simple solvable problem.

## 4. Methodology

Before discussing the proposed methodology, we present the used abbreviations, notations and perspective description in this paper as shown in Table 1.

The proposed pose-aware facial expression recognition framework is presented in Fig 1, which consists of the following 5 major parts:

**Table 1. Notations and description.**

| | |
|---|---|
| HOG | Histogram of oriented gradient |
| PCA | Principal component analysis |
| SNNE | Stacked neural network ensemble |
| NB | Naive Bayes |
| KNN | K nearest neighbor |
| SVM | Support vector machine |
| LBP | Local binary pattern |
| SIFT | Scale invariant feature transformation |
| RaFD | Radboud faces database |
| BNNs | Binary neural networks |

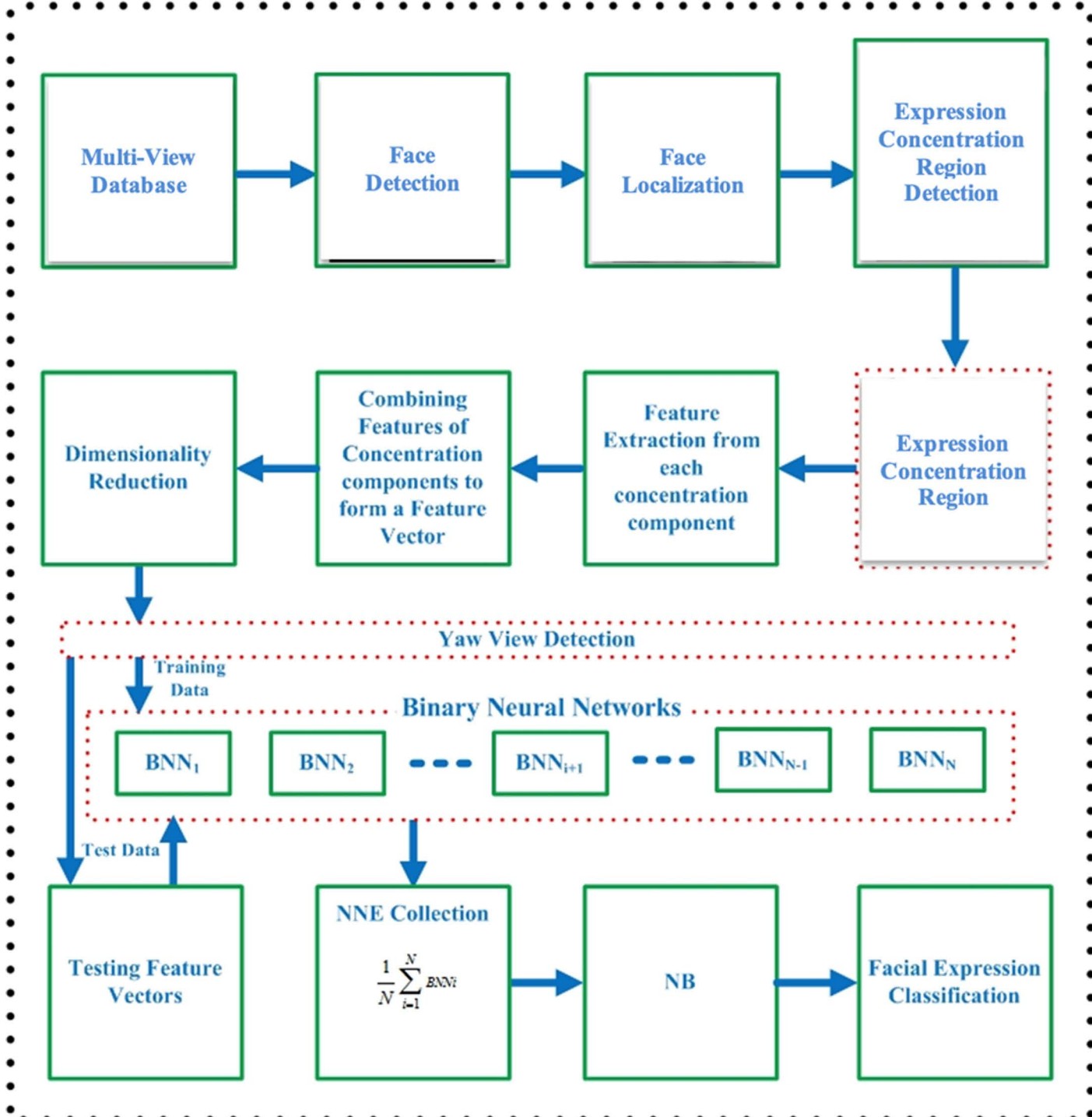

**Fig 1. Binary neural network-based stacking ensemble model for pose-aware facial expression recognition.**

## Pre-processing

This section largely focuses on methods for pre-processing images, such as noise reduction, normalization, threshold holding, sharpening, and cropping the face area. Therefore, it is crucial

to extract the facial picture that only shows the region of the face and has normalized intensity, is noise-free, and is consistent in size and shape. The localization and alignment of face pictures based solely on appearance is the initial stage in image pre-processing. The face alignment problem could be solved more effectively using the face alignment and detection techniques described in [23]. However, analyzing each new face picture computationally is expensive. Therefore, to extract the facial region from a picture, we employed Voila Jones' face detection technique. The size of the identified face region was adjusted to the mean of each batch of training poses. As a result, the face alignment stage only needs to conduct simple arithmetic computations briefly.

1. Feature extraction: In this phase, the method of face feature extraction from multiple-pose facial photos is examined. To depict a face picture, the feature extraction techniques PCA and HOG were both used. Principal component analysis was used to further analyze the retrieved feature vectors for dimensionality reduction, a similar method shown in [24].

2. Pose detection: This phase focuses on pose detection from multi-pose facial images. The major aim is to establish the relationship among the facial features of different poses using KNN classifier. In the KNN classifier, the response is the detection of pose from corresponding facial feature vector of the given facial image.

3. Ensemble classifier training: In a neural network ensemble, a binary neural network receives a one-dimensional feature vector with decreased dimensionality as input. Each SNNE consists of a collection of binary neural networks and a simple Bayes classifier. A group of binary neural networks trained to categorize each of the eight facial expressions were combined to create an SNNE. Each SNNE's binary output guarantees the existence or absence of a certain phrase. These ensembles' outcomes that are connected to the probability distribution span the SNNE.

4. Multi-pose facial expression recognition: This stage focuses on identifying many face expressions from testing samples. First, face characteristics from a sample image are retrieved, and then postures are found using a KNN classifier. The binary neural networks received the reduced feature vectors from the output of SNNEs after the postures had been determined. To reflect the most noticeable pose-ware facial expression, the output of SNNEs is connected to the NB classifier.

At last, we have observed that our experimentation requires to evaluate the types of classifiers.

- Firstly, the binary neural networks are trained to construct a SNNE. The output of binary neural networks across the SNNE was combined using majority voting approach. Although, it does not provide the evidence about any relationship between probability value for the presence of an individual expression with other SNNEs.

- A KNN classifier with city block distance measure and 10 nearest neighbors implemented to detect one of the possible five poses.

- A SVM classifier with different kernels was designed as a final predictor.

- A KNN with hamming distance implemented as a predictor to combine the decisions of SNNEs.

- The proposed NB classifier to combine the decision of SNNEs.

## Dataset preparation

The RaFD dataset contains 6840 images of 57 participants. The participants belong to Moroccans and Caucasians geographic regions. Out of 57 participants 38 are male participants and

19 female participants [24]. Each subject posed eight facial expressions (anger, happiness, surprise, sadness, fear, neutral, contempt, and disgust) with three gaze direction (left, right and front) and five head poses (-90°, +90°, -45°, +45°, 0°). The posed still images of each subject captured and digitized to 681 x 1024 pixels. Every subject in this database has three images of each facial expression against a pose.

The stacked ensemble models were trained using the RaFD multi-pose dataset. The facial region of each facial image was cropped using Voila Jones face detector as shown in Fig 2. The cropped facial region of each image varies in size and illumination. In order to create a consistent dataset, these photos are scaled to the average of the whole dataset of each posture. Finally, the picture histogram was equalized to normalize the impacts of lighting variance [25].

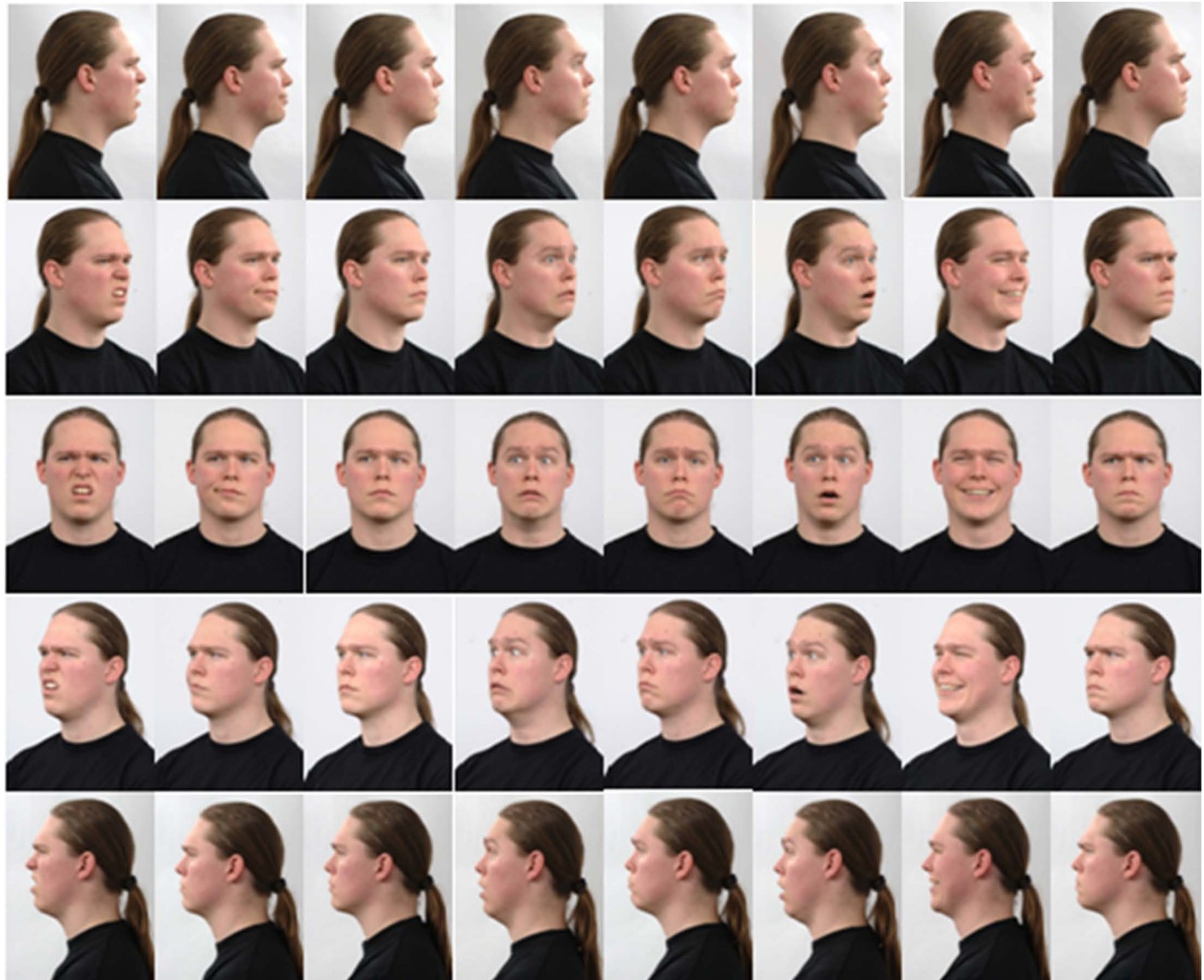

**Fig 2. Samples of the 40 facial images associated with the five poses and eight facial expressions of one subject in the RaFD (Row wise, 00, 450, Frontal, -00, -450) facial expression from left to right (anger, happiness, surprise, sadness, fear, neutral, contempt, disgust).**

## Neural network ensembles training

The principal component analysis and histogram-oriented gradients are used for feature extraction from sample images. Each sample image converted into a one-dimensional feature vector. This feature vector used to train the binary neural networks. For SNNEs construction first the subset expression specific data was randomly selected from training data, then another subset of other classes' samples is randomly selected to train the base-level classifier. A set of SNNEs is trained against each facial expression to recognize the one of the eight expressions. The output of stacked neural network ensembles depends on the predictions of base level classifiers, because the predictions of base-level classifiers are combined with majority voting to produce the output of SNNEs. During the training of SNNEs, if the performance of a base-level classifier is below 99%, that classifier was discarded. We used 0.3 learning rate, and 2000 maximum number of epochs, most of the binary neural networks (BNNs) converged before 50 epochs. Scaled conjugate gradient function was adopted as the neurons training function. The number of SNNEs in each stacked ensemble model was varied from 80 to 160, 240, 320 or 400 SNNEs. In case of 400 SNNE 50 SNNEs were trained against each expression. Finally, the decision about the number of SNNEs in each stacked ensemble model was made on the basis of expression accuracy achieved during training. The 50 most prominent features were selected to represent a facial image. The count of neurons in hidden layer was 10. Moreover, ½ of total training samples was randomly selected to train a base level classifier.

The binary neural networks trained using feature vectors of pose dependent facial expressions against all other expressions feature vectors. This model learns inter-expression variability in case of different head poses as described by Ekman [5,10]. Five pose-aware stacked ensemble models were trained using above-described parameters. The output of all SNNEs were concatenated to produce the meta input vector. The output of pose-aware stacked ensemble model is a 80, 160, 240, 320 or 400 element vector, representing the binary value about the presence of an expression. The output of a SNNE is determined using majority voting.

## Selection of parameters for ensemble classifier

Five types of stacked ensemble models were trained using previously mentioned parameters with respect to five poses (0°, −45°, +45°, −90°, +90°). Experimental results indicate that the accuracy of the 160, 240 and 400 SNNEs was higher than 80 and 320 SNNEs. The SNNEs count 160 give the best accuracy using Eigen vectors and achieving the best accuracy on frontal pose. These experiments showed that the performance of stacked ensemble model using SVM and KNN predictors is significantly lower than NB predictor. The optimal ensemble classifier structure (BNN training parameters, SNNE count, feature type and final predictor) for each pose are presented in Table 3.

## Pose detection

This part focuses on pose detection from multi-pose facial images. The major objective is to use KNN classifier for determining the relationship between the facial features of different poses. In the KNN classifier, the response is the pose of corresponding facial feature vector from the given facial pose. The KNN is the simplest classifier in machine learning [15]. It postponed the generalization task till the classification of sample data required. The algorithm predicts the class of sample data by computing the similarity between training set attributes and sample data attributes using some distance measure, and picks the K closest training samples. Assign the most common class among these training samples to the test sample. The

KNN was applied with varying value of K and distance measures. Finally, the KNN classifier with city block distance measure and K = 10 implemented to detect one of the possible five poses (−90°, −45°, 0°, +45°, +90°).

## Pose-aware facial expression recognition

We used NB classifier as a level-three classifier to combine the decisions of level-two classifiers in extended stacking ensemble model. The NB classifier determines the presence of one of the eight expressions in an unknown facial image.

### Naive Bays

Naive Bayes is a probability based simple classification technique; its performance is comparable to the most prominent classification techniques. The classifier predicts an unknown sample by defining a probabilistic relationship between the underlying features of trainings samples and test sample. It computes prior and posterior probabilities of sample features with respect to unknown sample while assuming no dependencies among all features. It assigns the class label to the unknown sample with the maximum likelihood of probability values. In practice NB performed significantly as compare to many state-of-the-art classification techniques.

## 5. Results

In this study, experiments are performed on RaFD multi-pose facial expression database. The detailed description of RaFD dataset is given in Table 2. The facial expression classification process carried out using RaFD database for each pose by varying the number of SNNEs (80, 160, 240, 320 and 400) with 10 BNNs. The NB, KNN and SVM classifiers are trained to predict the presence or absence of an expression. The results obtained from optimal ensemble classifier structure are presented in Table 4 along with accuracy on each expression. While complete evaluation results are presented in Tables 5–9.

Table 4 indicates the performance of stacked ensemble model with NB is superior using PCA as compare to HOG. However, results for expressions (neutral, sadness, fear and disgust) are very poor using HOG features. Therefore, accuracy differences using PCA and HOG features are very high on disgust, fear and neutral. The expression recognition accuracy difference is about 24.35% on disgust, about 11% on neutral and fear.

3 represents a deeper analysis of each expression on multi-pose expression classification using test dataset. The diagonal entries represent the correctly classified samples and off-diagonal entries specify the misclassified samples. The labels (AN stands for anger, HA for happiness, SU for surprise, SA for sadness, FE for fear, NE for neutral, CO for contempt and DI for disgust) are used for x-axis and y-axis. From these confusion matrices it is noticed

**Table 2. Description of multi-pose facial expression database.**

|  | Subject | No of Images | Pose | Relevant Expressions |
|---|---|---|---|---|
| Caucasians | 19 Female | 4680 | +90° +45° 0° −45° −90° | Anger Happiness Surprise |
|  | 20 Male |  |  | Sadness |
| Moroccans | 18 Male | 2160 |  | Fear Neutral Contempt Disgust |

**Table 3. Optimal classifier structure for ensemble classifier.**

| | PCA | | | | | HOG | | | | |
|---|---|---|---|---|---|---|---|---|---|---|
| Pose | −90° | −45° | 0° | ⁺45° | ⁺90° | −90° | −45° | 0° | ⁺45° | ⁺90° |
| NB | | | | | | | | | | |
| No of BNNs per SNNE | 10 | 10 | 10 | 10 | 10 | 10 | 10 | 10 | 10 | 10 |
| No of SNNEs | 80 | 320 | 160 | 400 | 240 | 400 | 400 | 400 | 240 | 320 |
| Accuracy | 80.00 | 88.29 | 90.00 | 87.07 | 79.76 | 70.24 | 82.20 | 86.59 | 84.39 | 66.83 |
| SVM | | | | | | | | | | |
| No of BNNs per SNNE | 10 | 10 | 10 | 10 | 10 | 10 | 10 | 10 | 10 | 10 |
| No of SNNEs | 400 | 400 | 400 | 400 | 240 | 400 | 400 | 400 | 400 | 400 |
| Accuracy | 72.68 | 85.85 | 88.29 | 81.22 | 72.20 | 62.93 | 80.00 | 84.15 | 79.51 | 64.15 |
| KNN | | | | | | | | | | |
| No of BNNs per SNNE | 10 | 10 | 10 | 10 | 10 | 10 | 10 | 10 | 10 | 10 |
| No of SNNEs | 400 | 160 | 160 | 80 | 240 | 160 | 240 | 400 | 400 | 160 |
| Accuracy | 77.56 | 87.80 | 88.78 | 86.10 | 78.78 | 63.66 | 80.00 | 86.34 | 81.95 | 63.90 |

**Table 4. Performance comparison of NB predictor with PCA and HOG features.**

| Expression | PCA | HOG |
|---|---|---|
| Anger | 78.18 | 80.36 |
| Happiness | 98.46 | 93.84 |
| Surprise | 91.73 | 86.52 |
| Sadness | 75.41 | 77.08 |
| Fear | 75.29 | 63.92 |
| Neutral | 75.23 | 63.80 |
| Contempt | 91.83 | 93.06 |
| Disgust | 86.95 | 62.60 |
| Average | 83.86 | 77.31 |

that the rate of misclassification is high between fear-surprise and fear-sadness expressions. Whereas neutral and disgust are the most confusing expressions. It represents the similar relationship between fear, sadness, and surprise expression on HOG features as presented in [26], where it is demonstrated that disgust is the most confusing expression as compare to others. The highest misclassification rate is between neutral-disgust and fear-surprise expressions, where 67 samples of fear are misclassified as surprise and 49 samples of neutral are misclassified as disgust. Moreover, these results demonstrates that among the eight multi-pose expressions, the expressions of surprise, contempt, and happiness are easier to be classified irrespective of pose-variations (Fig 3).

The experimental results broken down as pose wise from 0°, −45°, −90°, +45° to +90°. The detailed description of these results is presented in Tables 5–9. The illustrated results include the average accuracy along with performance of each expression with different combinations of SNNEs count and final predictor. The average results are the overall performances of the predictor, not the average of each column. The best accuracy is presented in bold face. Each SNNE is a set of 10 BNNs. The BNNs of each pose were trained using subset (specific facial pose images) feature vectors of each expression against similar facial pose feature vectors of other expressions. Therefore, the whole test data is used for the evaluation of five types of pose-aware extended stacking ensemble model.

**Table 5. Front pose facial expressions recognition (%) accuracy with NB, KNN, SVM predictors, and PCA, HOG features.**

| | 400 SNNEs | | | 320 SNNE | | | 240 SNNE | | | 160 SNNE | | | 80 SNNE | | |
|---|---|---|---|---|---|---|---|---|---|---|---|---|---|---|---|
| | NB | KNN | SVM | NB | KNN | SVM | NB | KNN | SVM | NB | KNN | SVM | NB | KNN | SVM |
| PCA | | | | | | | | | | | | | | | |
| Anger | 90.90 | 94.54 | 98.18 | 90.90 | 89.09 | 98.18 | 90.90 | 94.54 | 98.18 | 94.54 | 98.18 | 98.18 | 90.90 | 94.54 | 98.18 |
| Happiness | 98.07 | 98.07 | 98.07 | 98.07 | 98.07 | 96.15 | 98.07 | 98.07 | 96.15 | 98.07 | 98.07 | 96.15 | 98.07 | 98.07 | 96.15 |
| Surprise | 100 | 100 | 100 | 100 | 100 | 100 | 100 | 100 | 100 | 100 | 100 | 100 | 100 | 100 | 100 |
| Sadness | 75 | 70.83 | 81.25 | 72.91 | 75 | 81.25 | 75 | 64.58 | 81.25 | 75 | 68.75 | 81.25 | 72.91 | 81.25 | 81.25 |
| Fear | 92.15 | 90.19 | 80.39 | 92.15 | 92.15 | 78.43 | 92.15 | 90.19 | 78.13 | 92.15 | 92.15 | 76.47 | 92.15 | 78.43 | 76.47 |
| Neutral | 73.01 | 68.25 | 84.12 | 77.77 | 65.07 | 82.53 | 76.19 | 69.84 | 79.36 | 74.60 | 71.42 | 84.12 | 79.36 | 69.84 | 85.71 |
| Contempt | 95.91 | 93.87 | 93.87 | 95.91 | 93.87 | 89.79 | 95.91 | 93.87 | 91.83 | 95.91 | 93.87 | 89.79 | 95.91 | 93.87 | 81.63 |
| Disgust | 91.30 | 91.30 | 69.56 | 93.47 | 91.30 | 69.56 | 91.30 | 91.30 | 71.73 | 93.47 | 91.30 | 69.56 | 93.47 | 91.30 | 67.391 |
| Average | 89.02 | 87.8 | 88.29 | 89.76 | 87.32 | 87.07 | 89.51 | 87.32 | 87.07 | 90.00 | 88.78 | 87.07 | 90.00 | 87.80 | 86.10 |
| HOG | | | | | | | | | | | | | | | |
| Anger | 89.09 | 90.90 | 96.36 | 89.09 | 92.72 | 96.36 | 90.90 | 90.90 | 96.36 | 89.09 | 90.90 | 94.54 | 89.09 | 89.09 | 94.54 |
| Happiness | 92.30 | 92.30 | 98.07 | 92.30 | 94.23 | 100 | 92.30 | 92.30 | 98.07 | 92.30 | 94.23 | 98.07 | 88.46 | 90.38 | 98.07 |
| Surprise | 100 | 84.78 | 100 | 100 | 100 | 100 | 100 | 100 | 100 | 97.82 | 97.82 | 100 | 100 | 97.28 | 100 |
| Sadness | 85.41 | 81.25 | 70.83 | 85.41 | 77.08 | 72.91 | 83.33 | 75 | 70.83 | 85.41 | 79.16 | 75 | 87.50 | 79.16 | 70.83 |
| Fear | 78.43 | 96.07 | 62.74 | 78.43 | 86.27 | 68.62 | 76.47 | 84.31 | 68.62 | 74.50 | 80.39 | 62.74 | 76.47 | 74.50 | 58.82 |
| Neutral | 61.90 | 73.01 | 82.53 | 63.49 | 69.84 | 84.12 | 65.07 | 71.42 | 85.71 | 71.14 | 66.66 | 82.53 | 61.90 | 77.77 | 76.19 |
| Contempt | 100 | 100 | 100 | 100 | 100 | 100 | 100 | 100 | 100 | 100 | 100 | 95.91 | 100 | 100 | 93.87 |
| Disgust | 93.47 | 73.91 | 60.86 | 91.30 | 71.73 | 43.47 | 91.30 | 69.56 | 45.65 | 93.47 | 86.95 | 30.43 | 89.13 | 63.04 | 19.56 |
| Average | 86.59 | 86.34 | 84.15 | 86.59 | 86.10 | 83.66 | 86.59 | 85.12 | 83.66 | 85.12 | 86.34 | 80.49 | 85.61 | 83.90 | 77.07 |

**Table 6. Left (-45°) pose facial expressions recognition (%) accuracy with NB, KNN, SVM predictors, and PCA, HOG features.**

| | 400 SNNEs | | | 320 SNNE | | | 240 SNNE | | | 160 SNNE | | | 80 SNNE | | |
|---|---|---|---|---|---|---|---|---|---|---|---|---|---|---|---|
| | NB | KNN | SVM | NB | KNN | SVM | NB | KNN | SVM | NB | KNN | SVM | NB | KNN | SVM |
| PCA | | | | | | | | | | | | | | | |
| Anger | 87.27 | 80.00 | 96.36 | 89.09 | 83.63 | 96.36 | 87.27 | 81.81 | 96.36 | 89.09 | 85.45 | 96.36 | 87.27 | 83.63 | 96.36 |
| Happiness | 100 | 98.07 | 94.23 | 100 | 98.07 | 94.23 | 100 | 98.07 | 94.23 | 100 | 100 | 94.23 | 100 | 98.07 | 94.23 |
| Surprised | 100 | 95.65 | 100 | 100 | 100 | 100 | 100 | 100 | 100 | 100 | 100 | 100 | 100 | 100 | 100 |
| Sadness | 62.50 | 77.08 | 62.50 | 64.58 | 72.91 | 68.75 | 64.58 | 72.91 | 68.75 | 64.58 | 77.08 | 72.91 | 62.50 | 77.08 | 72.91 |
| Fear | 92.15 | 100 | 92.15 | 90.19 | 92.15 | 74.50 | 92.15 | 94.11 | 70.58 | 90.19 | 92.15 | 68.62 | 90.19 | 92.15 | 68.62 |
| Neutral | 88.88 | 60.31 | 87.30 | 90.47 | 61.90 | 85.71 | 80.95 | 68.25 | 84.12 | 90.47 | 88.88 | 84.12 | 77.77 | 84.12 | 82.53 |
| Contempt | 95.91 | 95.91 | 85.71 | 95.91 | 95.91 | 89.79 | 95.91 | 93.87 | 89.79 | 95.91 | 91.83 | 85.71 | 95.91 | 91.83 | 81.63 |
| Disgust | 76.08 | 86.95 | 65.21 | 73.91 | 86.95 | 67.39 | 84.78 | 86.95 | 65.21 | 71.73 | 65.21 | 65.21 | 89.13 | 65.21 | 63.04 |
| Average | 88.05 | 85.85 | 85.85 | 88.29 | 85.61 | 84.88 | 88.08 | 86.34 | 83.90 | 88.05 | 87.80 | 83.66 | 87.56 | 86.59 | 82.68 |
| HOG | | | | | | | | | | | | | | | |
| Anger | 89.09 | 76.36 | 94.54 | 87.27 | 69.09 | 94.54 | 90.90 | 72.72 | 94.54 | 89.09 | 89.09 | 96.36 | 81.81 | 87.27 | 96.36 |
| Happiness | 92.30 | 94.23 | 98.07 | 94.23 | 94.23 | 100 | 90.38 | 94.23 | 98.07 | 90.38 | 92.30 | 100 | 90.38 | 94.23 | 94.23 |
| Surprise | 91.30 | 97.82 | 100 | 91.30 | 97.82 | 100 | 91.30 | 95.65 | 97.82 | 89.13 | 97.82 | 100 | 91.30 | 95.65 | 100 |
| Sadness | 75.00 | 87.50 | 68.75 | 70.83 | 87.50 | 64.58 | 72.91 | 85.41 | 62.50 | 77.08 | 60.41 | 60.41 | 81.25 | 64.58 | 64.58 |
| Fear | 62.74 | 41.17 | 52.94 | 62.74 | 52.94 | 50.98 | 62.74 | 47.05 | 47.05 | 60.78 | 45.09 | 50.98 | 60.78 | 37.25 | 39.21 |
| Neutral | 84.12 | 95.07 | 85.71 | 80.95 | 76.19 | 77.77 | 84.12 | 74.60 | 80.95 | 84.12 | 74.60 | 76.19 | 82.53 | 77.77 | 73.01 |
| Contempt | 100 | 100 | 91.83 | 100 | 100 | 95.91 | 100 | 100 | 91.83 | 100 | 100 | 87.75 | 97.95 | 97.95 | 83.67 |
| Disgust | 60.86 | 73.91 | 43.47 | 60.86 | 63.04 | 43.47 | 60.86 | 73.91 | 41.30 | 60.86 | 65.21 | 36.95 | 60.86 | 65.21 | 21.73 |
| Average | 82.20 | 78.78 | 80.00 | 81.22 | 79.76 | 78.78 | 81.95 | 80.00 | 77.32 | 77.32 | 78.05 | 76.59 | 80.98 | 77.56 | 72.20 |

**Table 7. Left (-90°) pose facial expressions recognition (%) accuracy with NB, KNN, SVM predictors, and PCA, HOG features.**

| | 400 SNNEs | | | 320 SNNE | | | 240 SNNE | | | 160 SNNE | | | 80 SNNE | | |
|---|---|---|---|---|---|---|---|---|---|---|---|---|---|---|---|
| | NB | KNN | SVM | NB | KNN | SVM | NB | KNN | SVM | NB | KNN | SVM | NB | KNN | SVM |
| PCA | | | | | | | | | | | | | | | |
| Anger | 80.00 | 83.63 | 89.09 | 83.63 | 80.00 | 87.27 | 74.54 | 78.18 | 87.27 | 80.00 | 85.45 | 89.09 | 85.45 | 83.63 | 40.00 |
| Happiness | 96.15 | 88.46 | 98.07 | 96.15 | 90.38 | 98.07 | 94.23 | 90.38 | 98.07 | 96.15 | 94.23 | 94.23 | 94.23 | 86.53 | 98.07 |
| Surprise | 95.65 | 93.47 | 82.60 | 95.65 | 91.30 | 91.30 | 95.65 | 95.65 | 91.30 | 95.65 | 95.65 | 91.30 | 95.65 | 93.47 | 97.82 |
| Sad | 68.75 | 70.83 | 77.08 | 66.66 | 70.83 | 72.91 | 68.75 | 70.83 | 79.16 | 68.75 | 77.08 | 79.16 | 66.66 | 70.83 | 79.16 |
| Fear | 72.54 | 74.50 | 62.74 | 72.54 | 68.62 | 58.82 | 72.54 | 72.54 | 60.78 | 72.54 | 54.90 | 56.86 | 70.58 | 68.62 | 58.82 |
| Neutral | 63.49 | 58.73 | 68.25 | 63.49 | 52.38 | 66.66 | 66.66 | 53.96 | 68.25 | 63.49 | 58.73 | 66.66 | 66.66 | 66.66 | 76.19 |
| Contempt | 77.55 | 77.55 | 59.18 | 75.51 | 75.51 | 57.14 | 75.51 | 77.55 | 53.06 | 79.59 | 77.55 | 55.10 | 79.59 | 79.59 | 69.38 |
| Disgust | 84.78 | 78.26 | 41.30 | 82.60 | 76.08 | 39.13 | 82.60 | 80.43 | 34.78 | 82.60 | 71.73 | 34.78 | 84.78 | 41.30 | 41.30 |
| Average | 79.27 | 77.56 | 72.68 | 79.02 | 74.88 | 71.71 | 78.29 | 76.59 | 71.95 | 79.27 | 76.34 | 71.22 | 80.00 | 73.90 | 70.00 |
| HOG | | | | | | | | | | | | | | | |
| Anger | 81.81 | 61.81 | 94.54 | 85.45 | 58.18 | 94.54 | 83.63 | 70.90 | 92.72 | 87.27 | 74.54 | 63.63 | 81.81 | 83.63 | 76.36 |
| Happiness | 92.30 | 92.30 | 98.07 | 90.38 | 90.38 | 96.15 | 90.38 | 90.38 | 98.07 | 90.38 | 96.15 | 80.76 | 90.38 | 92.30 | 84.61 |
| Surprise | 82.60 | 78.26 | 91.30 | 80.43 | 80.43 | 93.47 | 78.26 | 78.26 | 93.47 | 82.60 | 80.43 | 80.43 | 78.26 | 78.26 | 76.08 |
| Sadness | 66.66 | 50.00 | 68.75 | 66.66 | 54.16 | 68.75 | 62.50 | 54.16 | 75.00 | 60.41 | 58.33 | 62.50 | 66.66 | 50.00 | 56.25 |
| Fear | 54.90 | 39.21 | 27.45 | 50.98 | 45.09 | 23.52 | 50.98 | 41.17 | 21.56 | 47.05 | 39.21 | 41.17 | 52.94 | 41.17 | 37.25 |
| Neutral | 53.96 | 38.09 | 41.26 | 55.55 | 33.33 | 28.57 | 55.55 | 22.22 | 23.80 | 55.55 | 46.03 | 50.79 | 53.96 | 30.15 | 36.50 |
| Contempt | 79.59 | 79.59 | 48.97 | 77.55 | 77.55 | 55.10 | 81.63 | 79.59 | 55.10 | 83.67 | 71.42 | 75.51 | 81.63 | 71.42 | 61.22 |
| Disgust | 52.17 | 54.34 | 34.78 | 52.17 | 58.69 | 34.78 | 56.52 | 52.17 | 21.73 | 56.52 | 45.65 | 43.47 | 45.65 | 36.95 | 39.13 |
| Average | 70.24 | 60.98 | 62.93 | 69.76 | 61.22 | 61.22 | 69.76 | 60.00 | 59.51 | 70.24 | 63.66 | 61.95 | 68.78 | 60.00 | 58.05 |

**Table 8. Right (+45°) pose facial expressions recognition (%) accuracy with NB, KNN, SVM predictors, and PCA, HOG features.**

| | 400 SNNEs | | | 320 SNNE | | | 240 SNNE | | | 160 SNNE | | | 80 SNNE | | |
|---|---|---|---|---|---|---|---|---|---|---|---|---|---|---|---|
| | NB | KNN | SVM | NB | KNN | SVM | NB | KNN | SVM | NB | KNN | SVM | NB | KNN | SVM |
| PCA | | | | | | | | | | | | | | | |
| Anger | 87.27 | 81.81 | 96.36 | 92.72 | 80 | 96.36 | 87.27 | 81.81 | 96.36 | 87.27 | 83.63 | 96.36 | 92.72 | 85.45 | 98.18 |
| Happiness | 100 | 100 | 96.15 | 100 | 100 | 96.15 | 100 | 100 | 98.07 | 100 | 100 | 96.15 | 100 | 100 | 96.15 |
| Surprise | 93.47 | 93.47 | 91.30 | 93.47 | 91.30 | 93.47 | 91.30 | 93.47 | 93.47 | 93.47 | 93.47 | 93.47 | 93.47 | 95.65 | 95.65 |
| Sadness | 85.41 | 85.41 | 70.83 | 81.25 | 85.41 | 72.91 | 83.33 | 85.41 | 72.91 | 85.41 | 85.41 | 72.91 | 77.08 | 85.41 | 72.91 |
| Fear | 82.35 | 84.31 | 82.35 | 82.35 | 86.27 | 80.39 | 82.35 | 84.31 | 70.58 | 82.35 | 84.31 | 70.58 | 82.35 | 82.35 | 64.70 |
| Neutral | 74.60 | 58.73 | 87.30 | 69.84 | 83.96 | 88.88 | 74.60 | 58.73 | 88.88 | 71.42 | 57.14 | 88.88 | 66.66 | 68.25 | 85.71 |
| Contempt | 89.79 | 91.83 | 73.46 | 91.83 | 91.83 | 73.46 | 91.83 | 91.83 | 73.46 | 91.83 | 93.87 | 73.46 | 91.83 | 91.83 | 73.46 |
| Disgust | 86.95 | 84.78 | 45.65 | 84.78 | 82.60 | 41.30 | 82.60 | 84.78 | 41.30 | 82.60 | 80.43 | 45.65 | 86.95 | 84.78 | 43.47 |
| Average | 87.07 | 84.15 | 81.22 | 86.59 | 82.93 | 81.22 | 86.34 | 84.15 | 80.24 | 86.34 | 83.90 | 80.49 | 85.85 | 86.10 | 79.51 |
| HOG | | | | | | | | | | | | | | | |
| Anger | 90.90 | 81.81 | 100 | 85.45 | 81.81 | 100 | 87.27 | 83.63 | 100 | 85.45 | 94.54 | 100 | 90.90 | 96.36 | 100 |
| Happiness | 96.15 | 96.15 | 98.07 | 96.15 | 96.15 | 94.23 | 96.15 | 96.15 | 94.23 | 96.15 | 96.15 | 94.23 | 96.15 | 96.15 | 94.23 |
| Surprise | 91.30 | 97.82 | 97.82 | 91.30 | 95.65 | 100 | 91.30 | 97.82 | 100 | 93.47 | 97.82 | 100 | 95.65 | 95.65 | 100 |
| Sadness | 85.41 | 81.25 | 70.83 | 85.41 | 83.33 | 70.83 | 87.50 | 91.66 | 68.75 | 83.33 | 77.08 | 68.75 | 75.00 | 79.16 | 70.83 |
| Fear | 64.70 | 66.66 | 52.94 | 62.74 | 58.82 | 50.98 | 68.62 | 50.98 | 49.01 | 64.70 | 52.94 | 49.01 | 64.70 | 56.86 | 52.94 |
| Neutral | 80.95 | 69.84 | 87.30 | 80.95 | 53.96 | 87.30 | 82.53 | 61.90 | 85.71 | 82.53 | 73.01 | 80.95 | 82.53 | 69.84 | 76.19 |
| Contempt | 97.95 | 100 | 93.87 | 97.95 | 95.91 | 91.83 | 97.95 | 97.95 | 89.79 | 97.95 | 95.91 | 83.67 | 97.95 | 97.95 | 83.67 |
| Disgust | 63.04 | 65.21 | 28.26 | 60.86 | 78.26 | 32.60 | 63.04 | 65.21 | 28.26 | 60.86 | 58.69 | 23.91 | 67.39 | 50.00 | 17.39 |
| Average | 83.90 | 81.95 | 79.51 | 82.68 | 79.51 | 79.27 | 84.39 | 80.00 | 77.80 | 83.17 | 80.73 | 75.85 | 83.90 | 80.24 | 75.12 |

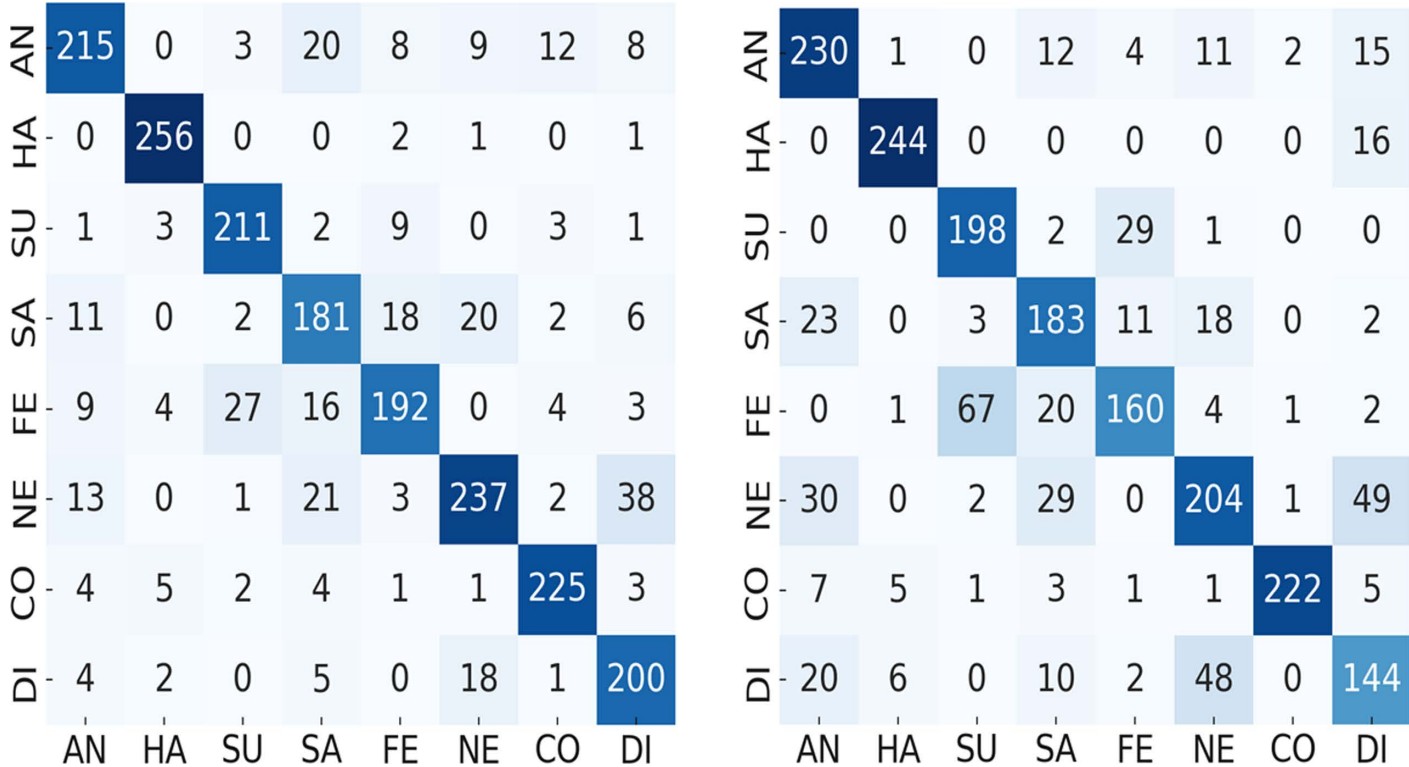

**Fig 3. Principal component analysis (left) and histogram of oriented gradient (right) expression confusion matrix.**

Table 5 shows the highest expression recognition accuracy on frontal pose with pose-aware ensemble model, with different SNNE count using PCA and HOG features. The results in this Table 5 indicate that the level of difficulty on frontal pose is significantly lower than non-frontal pose. Tables 6–9 present a significant drop in recognition accuracy as compare to the results presented in Table 5. The reason behind low recognition accuracies on non-frontal pose is the relatively less information available regarding facial structure and expression representation. These results indicate that pose-aware stacked ensemble model perform better with PCA and HOG descriptors. However, PCA achieved better performance as compare to HOG. The best average accuracies on frontal pose using PCA features are 90%, 88.78%, 88.29% and 86.59%, 86.34%, 84.15% using HOG features. The overall best recognition accuracy is 90% using Eigen features with 160 SNNEs and NB classifier.

The experimental results of all five types of pose-aware stacked ensemble models indicate that expression recognition accuracy for each expression varies with respect to different facial poses. It is observed that the most favorable facial pose is the frontal pose. In addition, among the three final predictors, NB achieves better recognition accuracy than KNN and SVM. This is most likely due to the fact that the NB learns using conditional probabilities of features derived from training data.

These experimental results indicate that the stacked ensemble model with NB predictor outperforms the other stacked ensemble models with KNN and SVM predictors respectively. Whereas, the combination of stacked ensemble model with KNN predictor outperforms the stacked ensemble model with SVM predictor trained using on both PCA and HOG features. The best average accuracies for stacked ensemble models with NB (90.00%), KNN (88.78%)

and SVM (88.78) using eigen vectors, illustrated in Table 5. In contrast, it can also observe that average classification accuracy of stacked ensemble model with SVM predictor is superior than with KNN predictor on facial pose +900 (presented in Table 9) when trained using HOG features. The expression recognition accuracy using HOG features is significantly lower as compare to PCA features.

The performance of pose-aware stacked ensemble model with the combination of NB and PCA is remarkably superior on frontal pose as compare to no-frontal pose. Its performance drops dramatically when applied on non-frontal facial images, as illustrated in Tables 6–9. This issue could also be pointed out by comparing the literature on multi-pose facial expression recognition [2,16] and [27], where higher recognition accuracy achieved on frontal pose facial images. It is observed that combination of all three final predictors with eigen vectors performed better than HOG features. The performance of three final predictors is remarkable on PCA. Moreover, considering the HOG features, the combination of HOG with NB outperformed the combination with KNN and SVM in five types pose-aware stacked ensemble model [28–35].

As Table 5 demonstrates the best expression recognition accuracy achieved with 160 SNNEs using eigen vectors. It also shows that pose-aware stacked ensemble model with NB predictor outperform the stacked ensemble models with KNN and SVM predictors respectively. We can so notice the stacked ensemble model trained using PCA features outperform the other stacked models trained with HOG features. Contrarily, the stacked ensemble model with KNN predictor performed significantly with lesser number of SNNEs. The experimented results presented in Table 6–9demonstrates that stacked ensemble approached better with 160 and 400 SNNEs as compare to other SNNEs count.

**Table 9. Right (+90°) pose facial expressions recognition (%) accuracy with NB, KNN, SVM predictors, and PCA, HOG features.**

|  | 400 SNNEs | | | 320 SNNE | | | 240 SNNE | | | 160 SNNE | | | 80 SNNE | | |
|---|---|---|---|---|---|---|---|---|---|---|---|---|---|---|---|
|  | NB | KNN | SVM | NB | KNN | SVM | NB | KNN | SVM | NB | KNN | SVM | NB | KNN | SVM |
| PCA | | | | | | | | | | | | | | | |
| Anger | 61.81 | 63.63 | 90.90 | 65.45 | 65.45 | 90.90 | 65.45 | 78.18 | 90.90 | 65.45 | 69.09 | 90.90 | 63.63 | 83.63 | 90.90 |
| Happiness | 100 | 100 | 96.15 | 100 | 100 | 94.23 | 100 | 92.30 | 94.23 | 100 | 100 | 94.23 | 100 | 100 | 94.23 |
| Surprise | 80.43 | 84.78 | 78.26 | 78.26 | 82.60 | 80.43 | 78.26 | 80.43 | 80.43 | 78.26 | 78.26 | 80.43 | 80.43 | 82.60 | 78.26 |
| Sadness | 83.33 | 81.25 | 70.83 | 83.33 | 83.33 | 70.83 | 81.25 | 79.16 | 70.83 | 83.33 | 83.33 | 68.75 | 83.33 | 68.75 | 70.83 |
| Fear | 64.70 | 66.66 | 52.94 | 62.74 | 62.74 | 47.05 | 68.62 | 70.58 | 49.01 | 62.74 | 68.62 | 45.09 | 62.74 | 60.78 | 43.13 |
| Neutral | 65.07 | 44.44 | 65.07 | 65.07 | 47.61 | 65.07 | 68.25 | 74.60 | 66.66 | 63.49 | 50.79 | 65.07 | 63.49 | 69.84 | 65.07 |
| Contempt | 93.87 | 97.95 | 63.26 | 95.91 | 93.87 | 61.22 | 93.87 | 91.83 | 63.26 | 95.91 | 75.51 | 57.14 | 95.91 | 95.91 | 57.14 |
| Disgust | 86.95 | 82.60 | 58.69 | 86.95 | 80.43 | 58.69 | 86.95 | 63.04 | 60.86 | 86.95 | 78.26 | 54.34 | 86.95 | 60.86 | 47.82 |
| Average | 78.78 | 76.34 | 72.20 | 79.02 | 75.85 | 71.22 | 79.76 | 78.78 | 72.20 | 78.78 | 74.63 | 69.76 | 78.78 | 77.80 | 68.78 |
| HOG | | | | | | | | | | | | | | | |
| Anger | 74.54 | 58.18 | 90.90 | 78.18 | 52.72 | 89.09 | 76.36 | 47.27 | 89.09 | 74.54 | 60.00 | 90.90 | 69.09 | 89.09 | 89.09 |
| Happiness | 96.15 | 96.15 | 96.15 | 96.15 | 96.15 | 96.15 | 96.15 | 96.15 | 94.23 | 94.23 | 96.15 | 96.15 | 94.23 | 98.07 | 98.07 |
| Surprise | 63.04 | 89.13 | 97.82 | 65.21 | 89.13 | 97.82 | 65.21 | 89.13 | 100 | 65.21 | 89.13 | 95.65 | 54.34 | 95.65 | 95.65 |
| Sadness | 64.58 | 72.91 | 60.41 | 66.66 | 79.16 | 58.33 | 62.50 | 77.08 | 62.50 | 60.41 | 72.91 | 64.58 | 54.16 | 56.25 | 56.25 |
| Fear | 49.01 | 31.37 | 31.37 | 50.98 | 31.37 | 25.49 | 47.05 | 33.33 | 25.49 | 49.01 | 37.25 | 23.52 | 56.86 | 15.68 | 15.68 |
| Neutral | 44.44 | 15.87 | 36.50 | 41.26 | 11.11 | 39.68 | 42.85 | 15.87 | 34.92 | 46.03 | 28.57 | 38.09 | 47.61 | 30.15 | 31.74 |
| Con tempt | 95.91 | 95.91 | 77.55 | 95.91 | 93.87 | 77.55 | 95.91 | 97.95 | 79.89 | 91.83 | 91.83 | 67.34 | 91.83 | 65.30 | 69.38 |
| Disgust | 43.47 | 45.65 | 26.08 | 43.47 | 52.17 | 23.91 | 39.13 | 52.17 | 21.73 | 36.95 | 45.65 | 19.56 | 36.95 | 4.34 | 15.21 |
| Average | 66.10 | 61.46 | 64.15 | 66.83 | 61.22 | 63.17 | 65.37 | 61.71 | 62.93 | 64.63 | 63.90 | 61.71 | 63.17 | 62.93 | 58.54 |

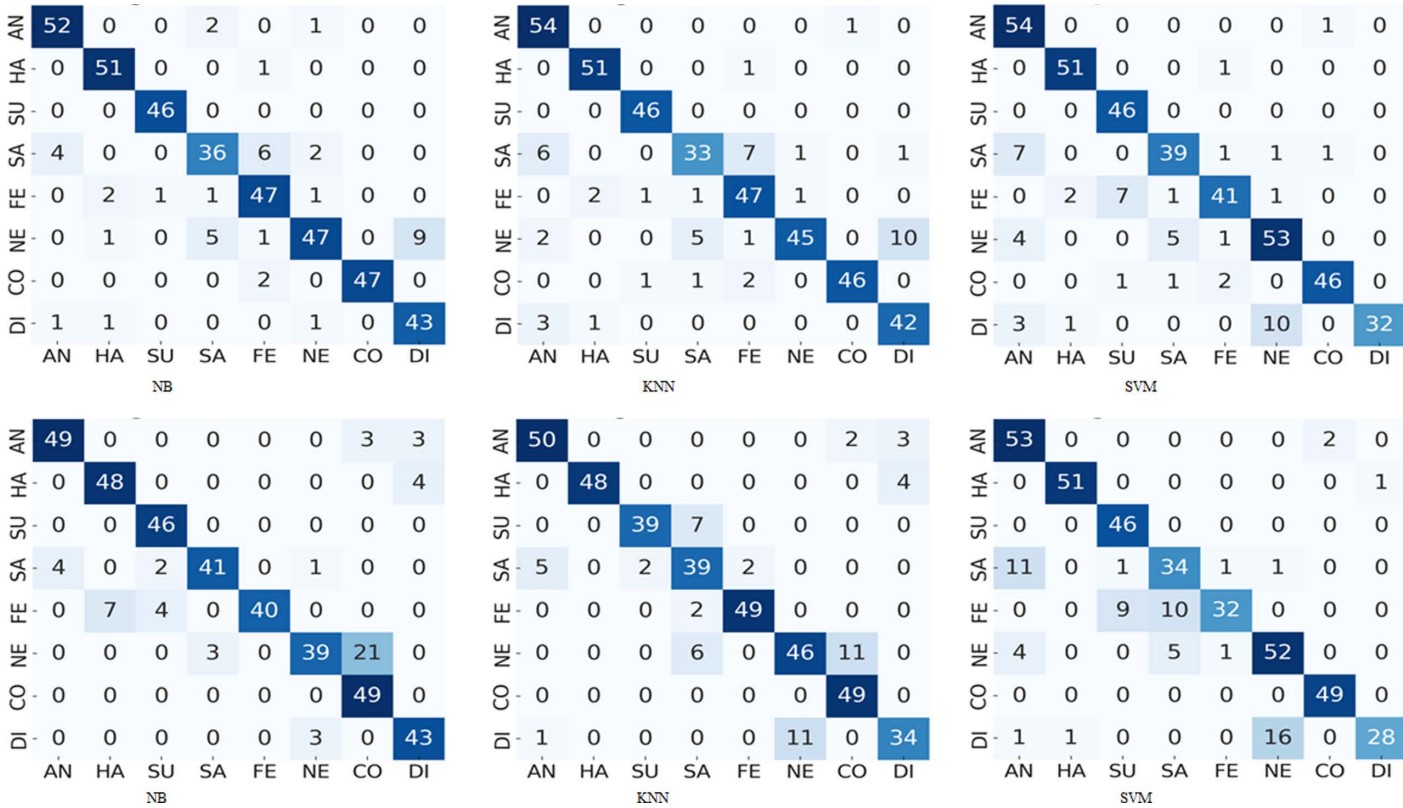

**Fig 4. The results of frontal pose confusion matrices comparing performance of NB, KNN and SVM on PCA and HOG features** : (a)-(c) confusion matrices correspond to PCA features, and (d)-(f) confusion matrices corresponds to HOG features.

Let us consider the confusion matrices illustrated in Figs 3–8 pertaining the best combination of stacked ensemble model with final predictor and feature extraction techniques. These figures show the resultant confusion matrices of five types of stacked ensemble models. The grey scale matrices are designed to visualize the results, where gray scale vary from black to white representing the intensity of values from high to low respectively. In contrast to diagonal values, which are difficult to perceive, diagonal values are easy to visualize. These confusion matrices represent the recognized facial expression on x-axis and facial image labels on y-axis, where each row represents the confusion intensity of each expression with respect to other expressions. The gray intensity in all confusion matrices represents the level of inter-expression similarity as well as dissimilarity. Fig 4 illustrates the level of confusion between different expression combinations with the best expression recognition accuracy of 90%. These confusion matrices illustrate that disgust and neutral are the most confused expressions. Moreover, from these confusion matrices we can see that, among the eight expressions, the expressions of happiness, surprise and contempt are easier to recognize than anger, sadness, fear, neutral and disgust. Considering the best performances each pose-aware stacked ensemble model, it can be notice that the happiness expression easy to recognize as compare to other expressions. Whereas, in the best performing stacked ensemble model, the expression surprise has the superior classification accuracy (100%).

To envision the confusion between eight expressions, we presented the confusion matrices of five types of pose-aware stacked ensemble models in Figs 4–8. These confusion matrices indicate that the expressions of surprise, contempt, and happiness are easier to recognize than

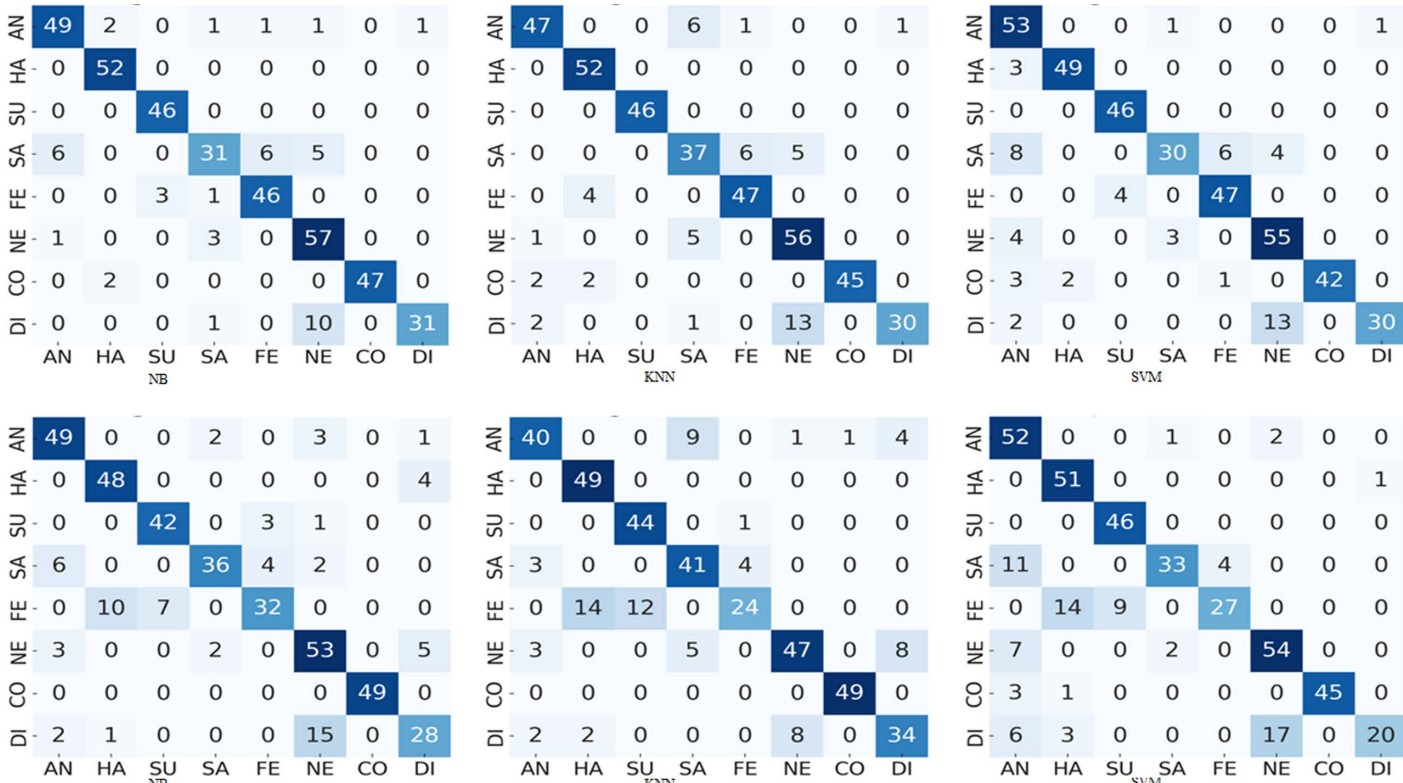

**Fig 5. The results of pose -45° yaw pose confusion matrices comparing performance of NB, KNN and SVM on PCA and HOG features** : (a)-(c) confusion matrices correspond to PCA features, and (d)-(f) **confusion matrices corresponds to HOG features.**

other expressions. It happens due the large muscular deformation in representing these three facial expressions as compare to other facial expressions. The expression surprise is the easiest one to recognize and disgust is the most confused. Moreover, these figures illustrate that disgust, fear and sadness expressions are difficulty to recognize due to confusion rate. Next to neutral and disgust, surprise and fear are the most confused expressions in non-frontal pose. This confusion may attribute to the similar low muscle deformation.

## 6. Discussion

During the training of SNNEs it is observed that stacked ensemble model consistently experienced higher accuracy on frontal pose as compare to non-frontal pose. It was also evident from the experimental results of Mostafa et al. [2] and Wenming [16], where different facial databases are used to train and test the expression recognition model. The expression recognition accuracies demonstrates that prediction rate of all expressions is not equal. For example, by observing human recognition accuracy for RaFD from [24], it can be noticed that surprise and happiness have the highest expression recognition accuracy as compare to other expressions. These findings strengthen the results of proposed ensemble model. It can also be noticed that fear and sadness expressions have lowest recognition accuracy and most confused expressions. It witnessed about the stability of stacked ensemble model.

It has been noticed that using the whole dataset for training and testing of classifiers, irrespective of pose variations, the performance of stacked ensemble model degrades severely. Table 10 demonstrates the expression recognition accuracies on multi-pose databases, where

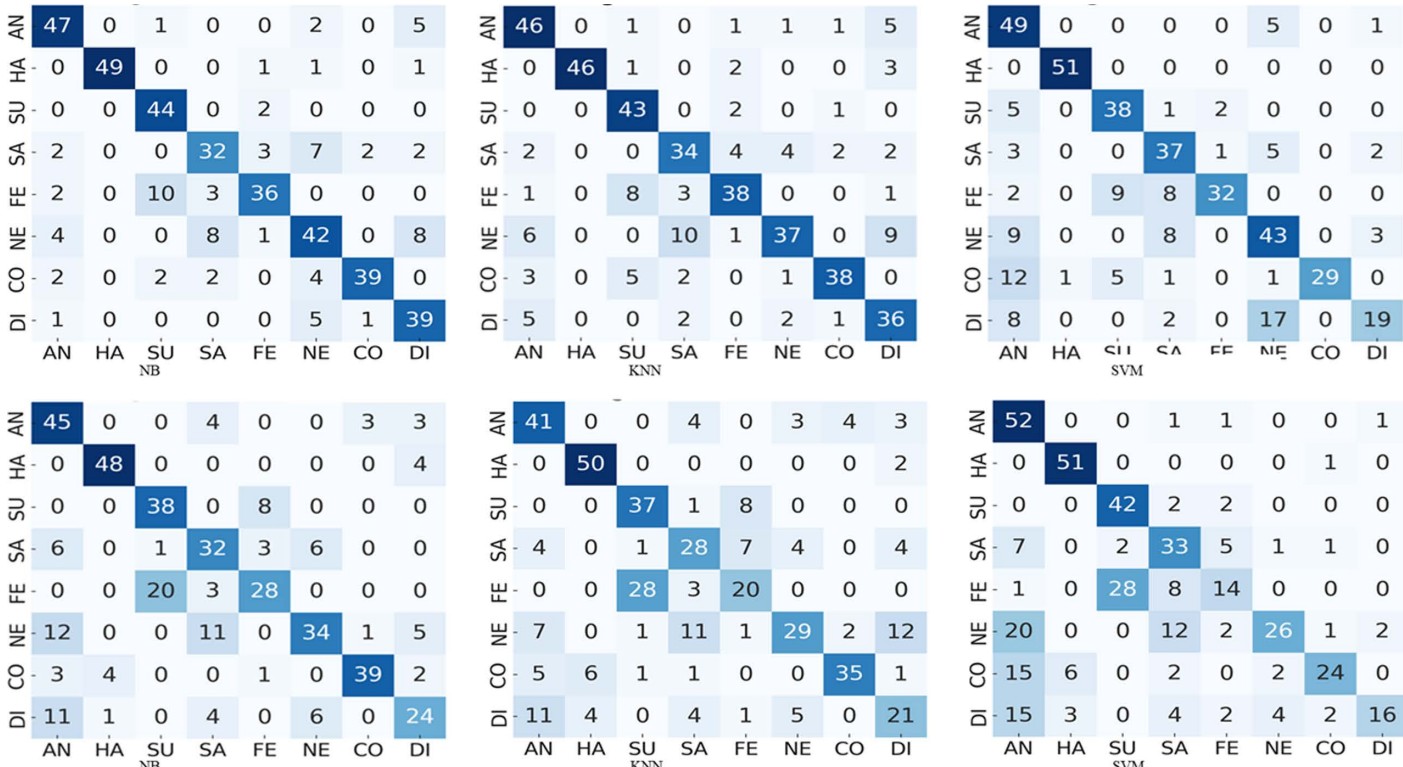

**Fig 6. The results of pose +45° yaw pose confusion matrices comparing performance of NB, KNN and SVM on PCA and HOG features:** (a)-(c) confusion matrices correspond to PCA features, and (d)-(f) confusion matrices corresponds to HOG features.

several datasets were utilized to train and evaluate expression recognition models These findings show that the accuracy of expression recognition models is quite poor when compared to using pose-specific datasets for training and testing. Again, this variation in expression recognition accuracy strengthens the evidence about the generalization of stacked ensemble model.

The most similar work to the proposed technique is presented in [41], where KEF dataset is used with five facial views (+90°, −90°, +45°, −45°, 0°). The best performance of proposed technique is 87% on −90° and 88% on −45° facial images. We can say that proposed approach performed slightly lower as compare to the work presented in [41]. Contrarily there is a huge difference in computational cost of both techniques. An evocative comparison of the results using RaFD with other databases used in literature is actually not possible. The reason behind that, up to the best of our knowledge, the existing work has not reported any results using multi-pose RaFD database. Instead, used only frontal pose of RaFD or BU-3DFE and Multi-PIE database. The results from other techniques using RaFD database listed in Table 11 for comparison.

The most noticeable work about the level of difficulty in recognizing the multi-pose facial expression is presented in [2]. The authors trained the expression recognition model on BU-3DFE dataset and evaluated its performance on five different multi-pose and front pose facial expression datasets (SPEW, RaFD, JAFFE, KDFE). Another similar work is presented in [27], where BU-3DFE dataset was used to train the classifier and Multi-PIE dataset was used for the testing of recognition model. Another interesting work is presented in [39]. Demonstrate the effectiveness of the Bayesian networks to capture the posed and spontaneous spatial features for gender and expression recognition. Table 10 illustrates most significant

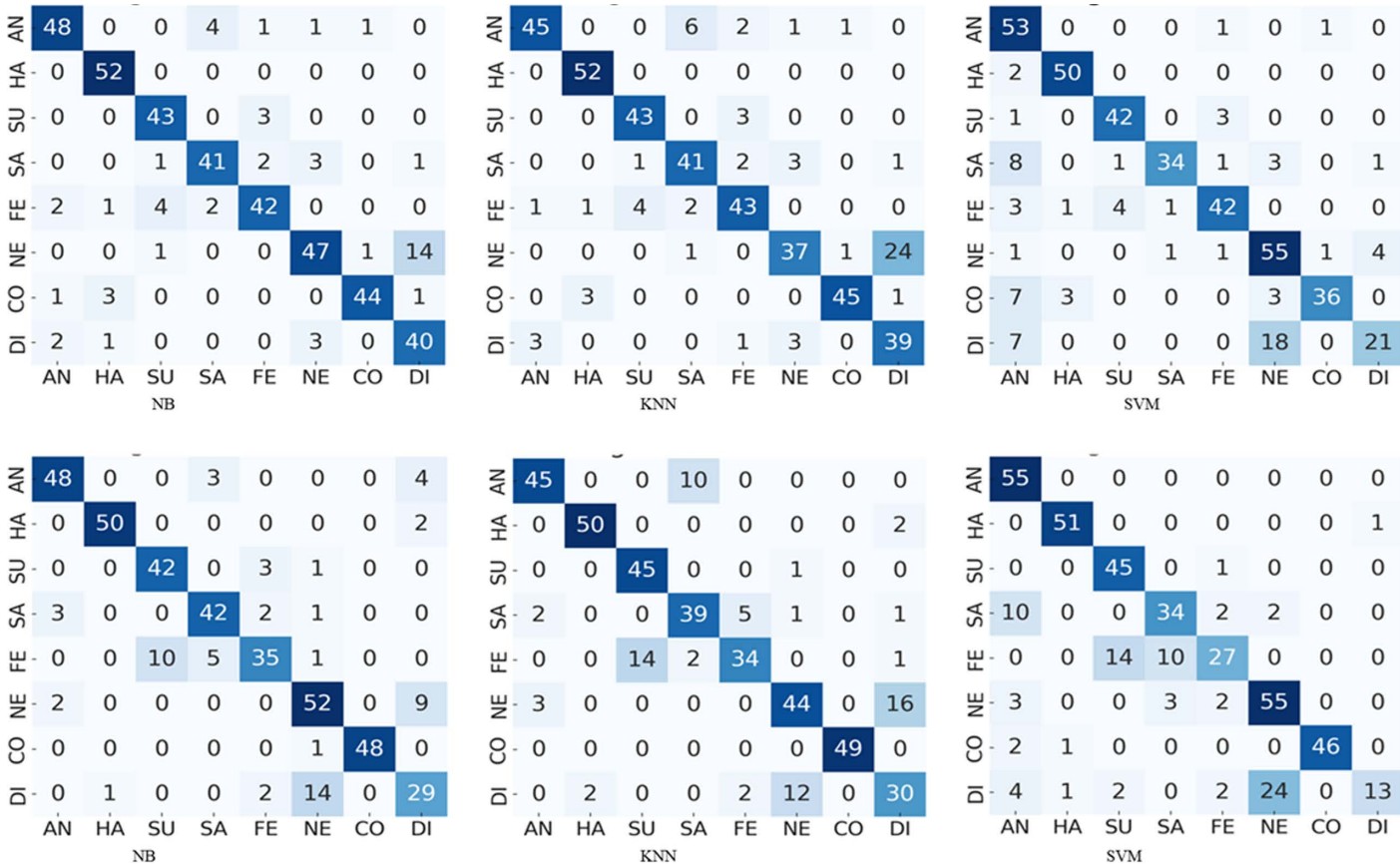

**Fig 7. The results of pose -90° confusion matrices comparing performance of NB, KNN and SVM on PCA and HOG featur**es, (a)-(c) confusion matrices correspond to PCA features, and (d)-(f) co**nfusion matrices** corresponds to HOG features.

results which are comparable to the performances achieved with pose-aware stacked ensemble model. The difference in performance of proposed ensemble model and other models is due to the difference in number of facial expressions and multi-pose expression representation. The work presented in [42] and [43] used only frontal pose images while achieving 98.1% and 95.6% accuracy with six and seven facial expressions respectively.

## 7. Conclusions

In this paper, we proposed the pose-aware stacked ensemble model for learning the facial expressions discrimination from multi-pose facial images. This ensemble approach is experimented on RaFD facial expression database to evaluate the recognition performance using PCA and HOG features. The experimental result shows that the proposed method achieves competitive recognition performance compared with the state-of-the-art methods. Further, the NB classifier through the incorporation of naive Bayes and Bernoulli distribution is a valuable combination where the outputs of SNNEs are binary. We would expect that the use of Bernoulli distribution would increase the capability of classifier to differentiate between expressions. The significant contribution of this research is the development of pose-aware ensemble model for multi-pose facial expression recognition. The introduction of the combination of pose detection and then pose specific facial expression recognition ensemble model is entirely a novel structure that gains the advantage pose dependent expression recognition.

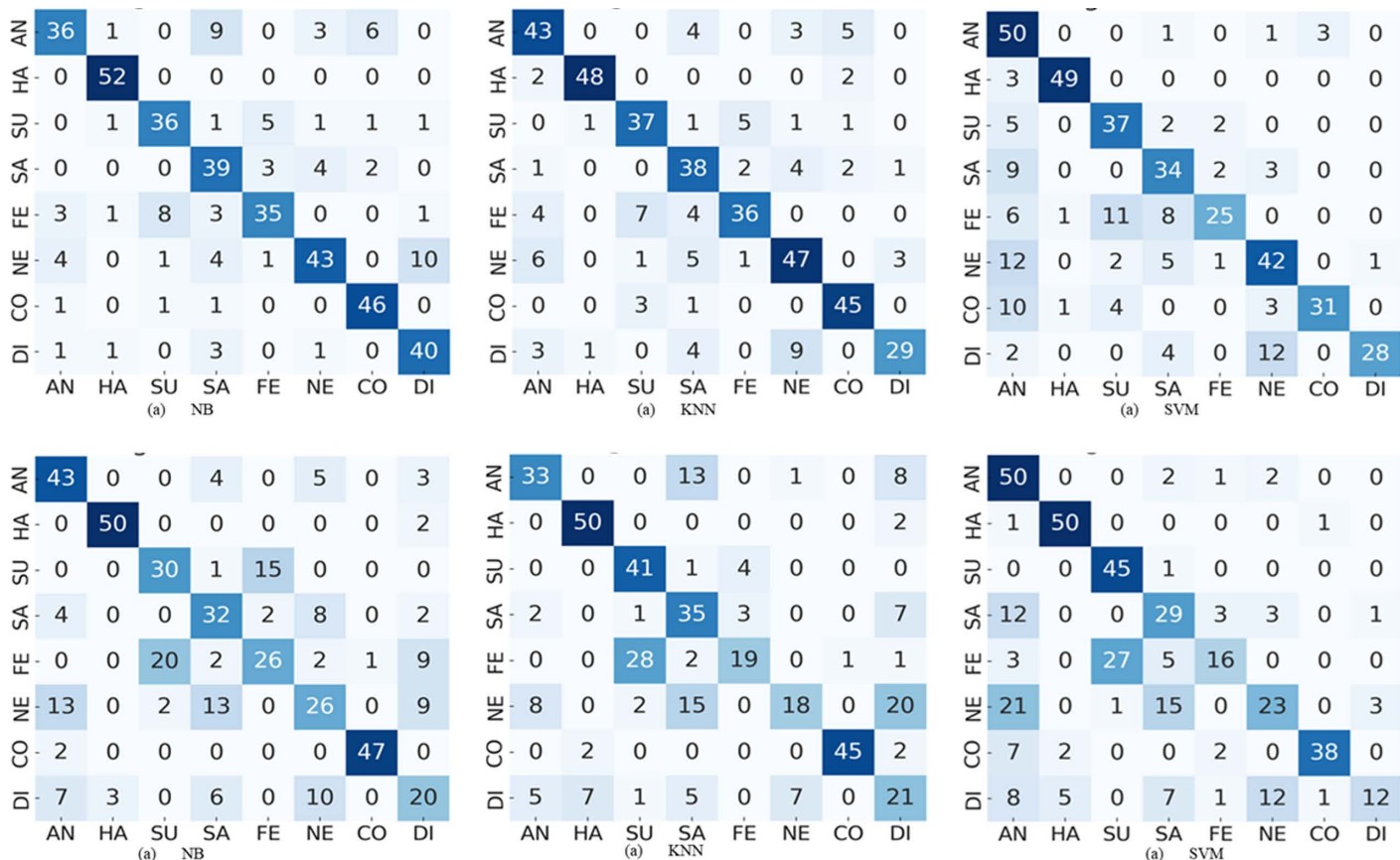

**Fig 8. The results of pose +90° confusion matrices comparing performance of stacked ensemble model of type 1 combined with NB, KNN and SVM on PCA and HOG features:** (a)-(c) confusion matrices correspond to PCA features, and (d)-(f) confusion matrices corresponds to HOG features.

**Table 10. Existing work on multi-pose facial expression recognition.**

| Reference | Classifier | Dataset | Accuracy | Remarks |
|---|---|---|---|---|
| 2014 [2] | Random Forest | RadBoud BU-3DFE | 78.91 | Trained classifiers using BU-3DFE and tested on multiple facial expression databases. |
| 2014 [27] | GSRRR | BU-3DFE | 78.90 | Synthesized the multi-pose facial feature vectors and combined to form single pose features. |
| 2014 [27] | GSRRR | Multi-PIE | 81.74 | |
| 2015 [36] | DS-GPLVM | Multi-PIE | 90.60 | Presented pose invariant feature extraction and classification technique |
| 2011 [37] | ANN | RadBoud | 90.0 | Presented performance of system described in [38] |
| 2012 [15] | SVM | BU-3DFE | 77.67 | Compared performance of LGBP features with different variants of LBP |
| 2012 [15] | SVM | Multi-PIE | 80.40 | |
| 2015 [39] | NB, SVM | USTC-NVIE | 85.51 | Presented a new method to recognize posed and spontaneous expressions through modeling their spatial patterns. |
| 2015 [39] | NB, SVM | SPOS | 74.79 | |
| 2015 [40] | SVM | CK$^+$ | 94.7 | Employed active shape model to extract different dynamic face regions. |
| 2023 [41] | CNN | KDEF | 88.8 93.2 | 88.8 with + -90° and 93.2 with + - 45° applied deep learning approaches |
| Proposed | NBSNNE450 | RaFD | 83.86 | PCA features |
| Proposed | NBSNNE450 | RaFD | 77.31 | HOG features |

**Table 11. Classification accuracy of existing methods which used RadBoud Faces Database.**

| Reference | Training Database | Test Database | Accuracy |
|---|---|---|---|
| 2014 [2] | BU-3DFE | RadBoud | 55.03 |
| 2011[37] | RadBoud | RadBoud | 90.00 |
| 2015[42] | RadBoud | RadBoud | 98.10 |
| 2013[43] | RadBoud | RadBoud | 95.60 |

However, SNNEs at meta level provide high generality by introducing a pool of binary neural networks for extended stacked ensemble learning. The binary output of base level classifier as well the meta level classifier enables the final predictor to accurately predict the on the eight facial expressions. Therefore, it is very difficult for a classifier to make efficiently recognize the facial expressions in case of varying pose and facial appearance. The results obtained using pose-aware stacked ensemble model are significantly comparable to the performance of multi-pose facial expression recognition techniques presented in literature [2,37,42,43]. Consequently, we observed that the combination of stacked ensemble model with NB outperform the SVM and KNN predictors.

## Acknowledgments

The authors extend their appreciation to the Department of Computer Science, GC Women University Sialkot, Pakistan for research support.

## Author contributions

**Conceptualization:** Muhammad Faheem Altaf, Khlood Shinan.

**Data curation:** Muhammad Waseem Iqbal, Khlood Shinan.

**Formal analysis:** M. Usman Ashraf, Fatmah Alanazi.

**Funding acquisition:** Khlood Shinan.

**Investigation:** Muhammad Waseem Iqbal, Fatmah Alanazi.

**Methodology:** Muhammad Faheem Altaf.

**Project administration:** Fatmah Alanazi.

**Resources:** Hanan E. Alhazmi, Fatmah Alanazi.

**Software:** Ghulam Ali.

**Validation:** Hanan E. Alhazmi.

**Visualization:** Hanan E. Alhazmi.

**Writing – original draft:** Ghulam Ali.

**Writing – review & editing:** Ghulam Ali.

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
