## [Decision Letter · Decision Letter 0]

23 Feb 2024

PONE-D-23-40162Neural Network-Based Ensemble Approach for Multi-View Facial Expression RecognitionPLOS ONE

Dear Dr. Ashraf,

Thank you for submitting your manuscript to PLOS ONE. After careful consideration, we feel that it has merit but does not fully meet PLOS ONE’s publication criteria as it currently stands. Therefore, we invite you to submit a revised version of the manuscript that addresses the points raised during the review process. Care has to be taken towards the related works considered. To compare the proposed method with the existing methods, do consider the latest developments/works.

We look forward to receiving your revised manuscript.

Kind regards,

Vijayalakshmi G V Mahesh, Ph.D

Academic Editor

PLOS ONE

Journal Requirements:

2. Thank you for submitting the above manuscript to PLOS ONE. During our internal evaluation of the manuscript, we found significant text overlap between your submission and previous work in the [introduction, conclusion, etc.].

Please revise the manuscript to rephrase the duplicated text, cite your sources, and provide details as to how the current manuscript advances on previous work. Please note that further consideration is dependent on the submission of a manuscript that addresses these concerns about the overlap in text with published work.

[If the overlap is with the authors’ own works: Moreover, upon submission, authors must confirm that the manuscript, or any related manuscript, is not currently under consideration or accepted elsewhere. If related work has been submitted to PLOS ONE or elsewhere, authors must include a copy with the submitted article. Reviewers will be asked to comment on the overlap between related submissions (http://journals.plos.org/plosone/s/submission-guidelines#loc-related-manuscripts).]

We will carefully review your manuscript upon resubmission and further consideration of the manuscript is dependent on the text overlap being addressed in full. Please ensure that your revision is thorough as failure to address the concerns to our satisfaction may result in your submission not being considered further.

5. We note that Figures 1 and 2 includes an image of a [patient / participant / in the study]. 

Reviewers' comments:

Reviewer's Responses to Questions

**Comments to the Author**

1. Is the manuscript technically sound, and do the data support the conclusions?

Reviewer #1: Yes

Reviewer #2: No

2. Has the statistical analysis been performed appropriately and rigorously? 

Reviewer #1: Yes

Reviewer #2: No

3. Have the authors made all data underlying the findings in their manuscript fully available?

Reviewer #1: Yes

Reviewer #2: Yes

4. Is the manuscript presented in an intelligible fashion and written in standard English?

Reviewer #1: Yes

Reviewer #2: No

5. Review Comments to the Author

Reviewer #1: 1.Try to put some digitizer results of the most important in this paper in the abstract. The abstract is the first image of a reader that can catch their eye or not.

2. "According to Ekman, the facial region around the eyes and mouth contains more information about

action units as compare to other regions of face." I recommend authors to add reference to this sentence.

3. Check the entire paper for abbreviations. For example, What is LBP, authors must write full name first and use the abbreviated term when it appeared next time.

4. Text inside the Figures is not clear. Kindly redraw or adjust the figures to maintain the quality.

5. Some of the relevant works on FER, and its applications in various areas are missing from the introduction, discussions, and references. The authors should add more relevant works. some of them are suggested below:

a) https://doi.org/10.1109/TIM.2023.3243661

b) https://doi.org/10.1109/TAFFC.2022.3208309

c) https://doi.org/10.1109/TIM.2023.3314815

d) https://doi.org/10.1109/TIM.2020.3031835

6. To check the generalizability authors must evaluate with another dataset (in-the-wild dataset) like RAFDB or FER2013.

7. It is observed that authors are not compared their proposed approach with recent FER studies. Suggesting authors to compare the performance of recent existing methods (Some approaches are suggested in comment no 5).

Reviewer #2: The manuscript lacks novelty. The authors have presented a study that lacks motivation. Why is there a need for a new method when we can achieve more accurate results (>90%) using models developed a decade ago? They have not justified its necessity. Moreover, the overall presentation needs improvement. The weakest section is the results, which require significant enhancement. Why PCA? More advanced versions are now available. These are the questions that need to be addressed.

6. PLOS authors have the option to publish the peer review history of their article (what does this mean? ). If published, this will include your full peer review and any attached files.

**Do you want your identity to be public for this peer review?** For information about this choice, including consent withdrawal, please see our Privacy Policy .

Reviewer #1: **Yes: ** Mohan Karnati

Reviewer #2: No

---

## [Author Response · Author response to Decision Letter 0]

26 Apr 2024

Reviewer 1

Comment 1: Try to put some digitizer results of the most important in this paper in the abstract. The abstract is the first image of a reader that can catch their eye or not.

Response: Thank you for the suggestion to include digitizer results in the abstract. We have revised the abstract to highlight key findings.

Comment 2: According to Ekman, the facial region around the eyes and mouth contains more information about units as compare to other regions of face." I recommend authors to add reference to this sentence.

Response: Thank you to highlight the importance of the reference to support the statement about the facial region around the eyes and mouth contain more information about facial expression, as per Ekman's research. We have added the appropriate reference of Ekman's article to substantiate this claim in the manuscript.

Comment 3: Check the entire paper for abbreviations. For example, What is LBP, authors must write full name first and use the abbreviated term when it appeared next time.

Response: Thank you to point out the need for consistency to define the abbreviations throughout the paper. We have reviewed the manuscript and ensured that all abbreviations, including LBP, are fully defined at their first appearance and used consistently thereafter.

Comment 4: Text inside the Figures is not clear. Kindly redraw or adjust the figures to maintain the quality.

Response: Thank you to bring our attention towards the clarity of text inside the figures. We have redrawn and adjusted Figure 3, Figure 4, Figure 5, Figure 6, Figure 7, and Figure 8 to enhance the quality, now all the text is clear and legible.

Comment 5: Some of the relevant works on FER, and its applications in various areas are missing from the introduction, discussions, and references. The authors should add more relevant works. some of them are suggested below:

Response: Thank you for the valuable feedback to include the relevant works on Facial Expression Recognition and its applications. We have updated the literature review section and discussed the latest research on facial expression recognition. Also, we have compared our proposed approach with the latest research in facial expression recognition and included the comparison in Table 8.

Comment 6: To check the generalizability authors must evaluate with another dataset (in-the-wild dataset) like RAFDB or FER2013.

Response: Thank you for your suggestion to evaluate the proposed technique on in-the-wild datasets like RAFDB or FER2013 to assess generalizability of the technique. The primary focus of this research is on the development of a pose-aware facial expression recognition technique with the K nearest neighbor for pose detection and a neural network-based extended stacking ensemble model for classification of pose-aware facial expressions. The proposed methodology was specifically designed and optimized for the Radboud faces database with multiple views. While we understand the importance of cross-dataset evaluation, our approach aims to demonstrate the efficacy and advancement within the context of the Radboud faces database, to achieve significant performance compared to state-of-the-art technique while considering the pose aware constraint.

Comment 7: It is observed that authors are not compared their proposed approach with recent FER studies. Suggesting authors to compare the performance of recent existing methods (Some approaches are suggested in comment no 5).

Response: Thank you to point out the importance of comparison of proposed approach with recent Facial Expression Recognition studies. We have conducted a performance comparison of proposed technique with the KDEF dataset to present its effectiveness. However, comparison with other datasets presents challenges due to the pose variations inherent in our methodology. We believe that our study's primary contribution lies in addressing pose-aware facial expression recognition, and the KDEF dataset serves as a relevant benchmark for this purpose.

Reviewer 2

Comment 1: The manuscript lacks novelty. The authors have presented a study that lacks motivation. Why is there a need for a new method when we can achieve more accurate results (>90%) using models developed a decade ago? They have not justified its necessity.

Response: Thank you for raising concerns regarding the novelty and motivation of this study. While it may be true that models developed a decade ago can achieve high accuracy, our study aims to address specific challenges in pose-aware facial expression recognition. Our proposed technique extends the stacking ensemble model to incorporate pose detection, which is a significant advancement in adapting facial expression recognition to pose variations. The novelty of this study is the ability to effectively handle pose variations, that enhance the robustness and applicability of facial expression recognition systems. We believe that this approach fills a gap in the literature regarding pose variation. The proposed techniques provide a specialized solution to pose-aware facial expression recognition, thereby justify the development of our new method.

Comment 2: Moreover, the overall presentation needs improvement. The weakest section is the results, which require significant enhancement. Why PCA? More advanced versions are now available. These are the questions that need to be addressed.

Response: Thank you for your feedback on the choice of feature extraction techniques in this research. We employed PCA and HOG feature extraction techniques independently to compare the effectiveness of two different techniques that capture different aspects of facial expressions and poses. The reason behind using the PCA is that it is an established technique for feature reduction and has proven to be one of the best performing methods for facial expression recognition, because it focuses on global variations in facial features. On the other hand, HOG features capture the local texture and shape information, these property makes these techniques suitable to detect detail of facial features and poses. In this study on pose-aware facial expression recognition, we evaluated the effectiveness of both feature extraction techniques (PCA and HOG) features separately to assess their individual contributions to the recognition performance. This approach allowed us to understand the strengths and limitations of each feature extraction technique and determine their suitability for pose aware facial expression recognition.

---

## [Decision Letter · Decision Letter 1]

25 Jun 2024

PONE-D-23-40162R1Neural Network-Based Ensemble Approach for Multi-View Facial Expression RecognitionPLOS ONE

Dear Dr. Ashraf,

Thank you for submitting your manuscript to PLOS ONE. After careful consideration, we feel that it has merit but does not fully meet PLOS ONE’s publication criteria as it currently stands. Therefore, we invite you to submit a revised version of the manuscript that addresses the points raised during the review process.

We look forward to receiving your revised manuscript.

Kind regards,

Vijayalakshmi G V Mahesh, Ph.D

Academic Editor

PLOS ONE

Additional Editor Comments:

The article requires revision.

Do consider the concerns raised to improve the article.

Reviewers' comments:

Reviewer's Responses to Questions

**Comments to the Author**

1. If the authors have adequately addressed your comments raised in a previous round of review and you feel that this manuscript is now acceptable for publication, you may indicate that here to bypass the “Comments to the Author” section, enter your conflict of interest statement in the “Confidential to Editor” section, and submit your "Accept" recommendation.

Reviewer #1: All comments have been addressed

Reviewer #3: (No Response)

2. Is the manuscript technically sound, and do the data support the conclusions?

Reviewer #1: Yes

Reviewer #3: Partly

3. Has the statistical analysis been performed appropriately and rigorously? 

Reviewer #1: Yes

Reviewer #3: (No Response)

4. Have the authors made all data underlying the findings in their manuscript fully available?

Reviewer #1: Yes

Reviewer #3: (No Response)

5. Is the manuscript presented in an intelligible fashion and written in standard English?

Reviewer #1: Yes

Reviewer #3: (No Response)

6. Review Comments to the Author

Reviewer #1: Authors have addressed all the comments. Now the paper quality has been improved, I congratulate authors for their efforts.

Reviewer #3: (No Response)

7. PLOS authors have the option to publish the peer review history of their article (what does this mean? ). If published, this will include your full peer review and any attached files.

**Do you want your identity to be public for this peer review?** For information about this choice, including consent withdrawal, please see our Privacy Policy .

Reviewer #1: No

Reviewer #3: No

---

## [Author Response · Author response to Decision Letter 1]

30 Jun 2024

We are really grateful to all the reviewers for their precious time to review our work and giving constructive suggestions for the improvement. We paper is in third round, and all the reviewers accepted for the publication.

---

## [Decision Letter · Decision Letter 2]

13 Dec 2024

Neural Network-Based Ensemble Approach for Multi-View Facial Expression Recognition

PONE-D-23-40162R2

Dear Dr. Ashraf,

We’re pleased to inform you that your manuscript has been judged scientifically suitable for publication and will be formally accepted for publication once it meets all outstanding technical requirements.

Kind regards,

Vijayalakshmi G V Mahesh, Ph.D

Academic Editor

PLOS ONE

Reviewers' comments:

Reviewer's Responses to Questions

**Comments to the Author**

1. If the authors have adequately addressed your comments raised in a previous round of review and you feel that this manuscript is now acceptable for publication, you may indicate that here to bypass the “Comments to the Author” section, enter your conflict of interest statement in the “Confidential to Editor” section, and submit your "Accept" recommendation.

Reviewer #4: All comments have been addressed

Reviewer #5: All comments have been addressed

2. Is the manuscript technically sound, and do the data support the conclusions?

Reviewer #4: Yes

Reviewer #5: Yes

3. Has the statistical analysis been performed appropriately and rigorously? 

Reviewer #4: Yes

Reviewer #5: Yes

4. Have the authors made all data underlying the findings in their manuscript fully available?

Reviewer #4: Yes

Reviewer #5: Yes

5. Is the manuscript presented in an intelligible fashion and written in standard English?

Reviewer #4: Yes

Reviewer #5: Yes

6. Review Comments to the Author

Reviewer #4: In the abstract mentioned he Radboud faces database was used for stacked ensembles’ training and testing

purpose. but in the methodology mentioned RaFD multi-pose facial expression database , kindly update the correct database name

Reviewer #5: Recommended for publishing ,Good work , references are right , I can to published, aligment of the paper is good

7. PLOS authors have the option to publish the peer review history of their article (what does this mean? ). If published, this will include your full peer review and any attached files.

**Do you want your identity to be public for this peer review?** For information about this choice, including consent withdrawal, please see our Privacy Policy .

Reviewer #4: No

Reviewer #5: No

---

## [Editor Report · Acceptance letter]

PONE-D-23-40162R2

PLOS ONE

Dear Dr. Ashraf,

I'm pleased to inform you that your manuscript has been deemed suitable for publication in PLOS ONE. Congratulations! Your manuscript is now being handed over to our production team.

Kind regards,

on behalf of

Dr. Vijayalakshmi G V Mahesh

Academic Editor

PLOS ONE